# SEE-DPO: Self Entropy Enhanced Direct Preference Optimization

**Shivanshu Shekhar**                                                    *shekhar6@illinois.edu*
*Siebel School of Computing and Data Science*
*University of Illinois Urbana-Champaign*

**Shreyas Singh**                                                    *shreyas.singh@fractal.ai*
*Fractal AI Research*

**Tong Zhang**                                                    *tozhang@illinois.edu*
*Siebel School of Computing and Data Science*
*University of Illinois Urbana-Champaign*

**Reviewed on OpenReview:** *https://openreview.net/forum?id=xQbRFHfgGL*

## Abstract

Direct Preference Optimization (DPO) has been successfully used to align large language models (LLMs) according to human preferences, and more recently it has also been applied to improving the quality of text-to-image diffusion models. However, DPO-based methods such as SPO, Diffusion-DPO, and D3PO are highly susceptible to overfitting and reward hacking, especially when the generative model is optimized to fit out-of-distribution during prolonged training. To overcome these challenges and stabilize the training of diffusion models, we introduce a self-entropy regularization mechanism in reinforcement learning from human feedback. This enhancement improves DPO training by encouraging broader exploration and greater robustness. Our regularization technique effectively mitigates reward hacking, leading to improved stability and enhanced image quality across the latent space. Extensive experiments demonstrate that integrating human feedback with self-entropy regularization can significantly boost image diversity and specificity, achieving state-of-the-art results on key image generation metrics.

## 1 Introduction

Recent advancements in text-to-image generation models have achieved remarkable success Rombach et al. (2022); Podell et al. (2023); Saharia et al. (2022); Schneuing et al. (2024), enabling the creation of high-quality images that are both visually striking and semantically coherent. By leveraging large-scale image datasets, these models have demonstrated an unprecedented ability to generate images that align closely with the intended descriptions. This progress has garnered significant attention, highlighting the vast potential for practical applications in fields such as art, design, and content generation, while also raising important questions regarding the broader implications of such technology. Saharia et al. (2022); Schneuing et al. (2024); Dhariwal & Nichol (2021)

Text-to-image generation models are typically pretrained on vast datasets, which often contain low-quality contents, leading to the introduction of various deficiencies in the trained models Rombach et al. (2022). To mitigate these issues and improve the quality of generated outputs, additional post-training steps are usually required. Common approaches for post-training include Supervised Fine-Tuning (SFT) Brown et al. (2020a) and Reinforcement Learning with Human Feedback (RLHF) Christiano et al. (2023). RLHF has gained popularity in recent years due to its ability to improve generative model's ability using self generated data Bai et al. (2022). RLHF has been highly successful in aligning LLMs Rafailov et al. (2024b); Brown

et al. (2020b); OpenAI et al. (2024), and more recently it has demonstrated great potential for enhancing the generation quality of diffusion models Yang et al. (2024); Wallace et al. (2023); Black et al. (2024); Fan et al. (2023); Liang et al. (2024).

Standard RLHF algorithms rely on a reward model to evaluate output images and a learning algorithm, such as Proximal Policy Optimization (PPO) Schulman et al. (2017) or REINFORCE Zhang et al. (2020), to adjust the generative model based on these rewards. Reward models are often trained on human preference data Kirstain et al. (2023); Wu et al. (2023). More recently, Direct Preference Optimization (DPO) fine-tunes a large language model directly on preference data without reward modeling Rafailov et al. (2024b;a). The algorithm became popular because it yields competitive performance and is more stable to train than PPO, and has since been applied to diffusion models as well Wallace et al. (2023); Yang et al. (2024); Liang et al. (2024). Methods like D3PO Yang et al. (2024), SPO Liang et al. (2024) and Diffusion-DPO Wallace et al. (2023) adapt the DPO objective to train diffusion models. Some more recent studies showed that for large language models, online iterative DPO plus reward model learning, is superior to DPO Yuan et al. (2024); Xiong et al. (2024), and similar techniques can be applied to diffusion models as well. Online DPO algorithms consist of two key steps: model training and sample generation. At each timestep, the current model undergoes fine-tuning on the available dataset using DPO. Following this, new samples are generated from the updated model. These samples are subsequently evaluated and ranked by an existing reward model. The newly scored data is then appended to the existing dataset, ensuring continuous improvement and adaptation of the model. This is the approach we will employ in this paper.

However, for large language models, when we use online iterative DPO for multiple iterations, the generation model deteriorates significantly due to major shifts from the original distribution which leads to artificially high rewards and mode collapse. This phenomenon, referred to as reward hacking Xu et al. (2024), is a challenging problem that we have also observed in diffusion models. [Fig: 1]

A commonly used method to mitigate reward hacking is to include a KL regularization term in the objective function that enforces the trained language model to be close to the pretrained model (reference distribution) Christiano et al. (2023). We observe that this regularization, while mitigating the reward hacking problem, is insufficient to resolve it for diffusion models. We identify a key reason for this to be that pretrained diffusion model's generation is not diverse enough, and thus after RLHF, the trained model would have further reduced diversity. To address this problem, we propose to incorporate an additional self-entropy regularization to improve diversity. Reduced diversity means that the generation distribution is highly concentrated at a few points, causing the model to repeatedly output very similar images when prompted. Mathematically, we demonstrate that our approach flattens the reference distribution in the KL-divergence term. Therefore the regularization condition encourages the model to be more exploratory, effectively reducing overfitting and mitigating reward hacking. Moreover, our method achieves state-of-the-art performance across various metrics. Finally, we qualitatively show that our approach generates more diverse and aesthetically superior images compared to existing methods, which tend to produce high-quality results only for a limited subset of the input latent space for a given prompt (that is, lacks diversity). Our main contributions are as follows:

1. We introduce a novel self-entropy regularization term into the standard KL-regularized formulation of RLHF objective function. We show that this additional term can effectively improve RLHF results for diffusion models.

2. Our method is effective in mitigating reward hacking in DPO-based algorithms for diffusion models. When combined with the prior state-of-the-art methods D3PO, Diffusion-DPO and SPO we are able to obtain new state-of-the-art results on various image quality metrics.

3. We carry out empirical studies to demonstrate that this regularization technique encourages broader exploration of the solution space, reducing overfitting and preventing reward hacking. Moreover, we achieve enhanced stability and image quality across the latent space of the diffusion model, leading to more consistent outputs.

## 2 Related Works

**Diffusion Model:** Diffusion models have become state-of-the-art in image generation Rombach et al. (2022); Podell et al. (2023), with methods like Denoising Diffusion Probabilistic Models (DDPM) Ho et al. (2020) and Denoising Diffusion Implicit Models (DDIM) Song et al. (2022) significantly improving both image quality and generation speed. Text-to-image diffusion models, in particular, have enabled the creation of highly detailed and realistic digital art Rombach et al. (2022); Podell et al. (2023); Saharia et al. (2022). There has been an increased interest in adapting these models to specialized domains Li et al. (2023); Chi et al. (2024); Ajay et al. (2023); Ho et al. (2022), such as chemical structure generation Schneuing et al. (2024), where tuning the full model is computationally expensive.

Works like Hu et al. (2021); Lai et al. (2024); Luo et al. (2024) have introduced methods such as adapters and compositional approaches to combine multiple models, enhancing specificity, control, and output quality while lowering the computational needs for training. Additionally, classifier-free guidance (CFG) Ho & Salimans (2022) has streamlined the sampling process by removing the need for a separate classifier, instead relying on the model's predictions of both conditional and unconditional noise to improve quality and diversity. In contrast, classifier-based guidance Dhariwal & Nichol (2021), while providing more precise control, comes with higher computational costs due to the need for an external classifier.

**RLHF:** RLHF Christiano et al. (2023) has become a pivotal technique for aligning the outputs of foundational models with human preferences by incorporating human feedback during the training process. This method enables models to be fine-tuned to generate behaviorally aligned outputs, addressing issues like bias or harmful content. RLHF has been successfully applied across diverse domains, from developing policies for Atari games to training robotic systems Bai et al. (2022).

Notably, OpenAI's GPT series OpenAI et al. (2024) has showcased the impact of RLHF in enhancing language model performance, other parallel works have also shown a similar improvement across tasks Anthropic (2024); Google (2024); Touvron et al. (2023), particularly in improving coherence, relevance, and user experience. In parallel, Reinforcement Learning from AI Feedback (RLAIF) has been proposed as a cost-effective alternative, utilizing AI systems for feedback to scale model refinement. In this paper, we adopt RLAIF to generate high-quality images and achieve state-of-the-art performance across multiple evaluation metrics.

**DPO:** A key challenge in RLHF lies in designing an effective reward function, which typically requires a large, curated dataset for training in order to achieve competitive results Christiano et al. (2023). Even after constructing the reward model, it is vulnerable to manipulation by policies, especially for out-of-distribution (OOD) samples, leading to artificially inflated rewards. Additionally, training the main policy while simultaneously loading the reward model into memory introduces further computational complexity.

To address these issues, Rafailov et al. (2024b) introduced Direct Preference Optimization (DPO), a method that fine-tunes language models directly from user preferences, leveraging the correlation between reward functions and optimal policies. However, this approach does not directly extend to diffusion models, as DPO operates within the Contextual Bandit framework, which treats generation as a single-step process, whereas diffusion models require handling latent states over multiple timesteps, making training infeasible even with techniques like LoRA Hu et al. (2021). Diffusion-DPO Wallace et al. (2023) adapts this by optimizing an upper bound on the true loss function, D3PO Yang et al. (2024) models the diffusion process as a Markov Decision Process (MDP), using the action-value function instead of a traditional reward model at each generation step. This approach assumes that the chain generating the preferred response is best at every step. In contrast, SPO Liang et al. (2024) introduces training a reward model that can score samples across varying noise levels, allowing for effective optimization of the DPO objective throughout the diffusion process. This method relaxes the strict assumptions made by D3PO, providing a more flexible approach to diffusion modeling.

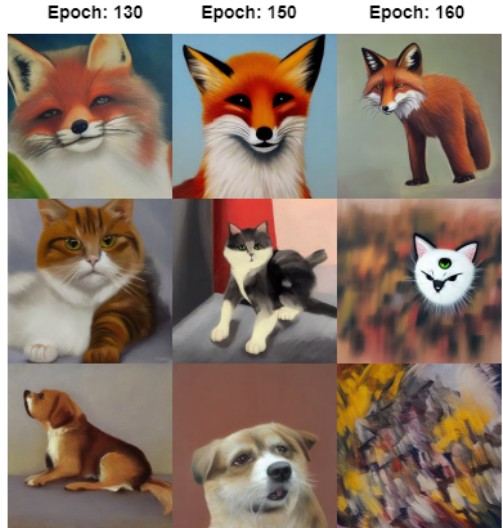 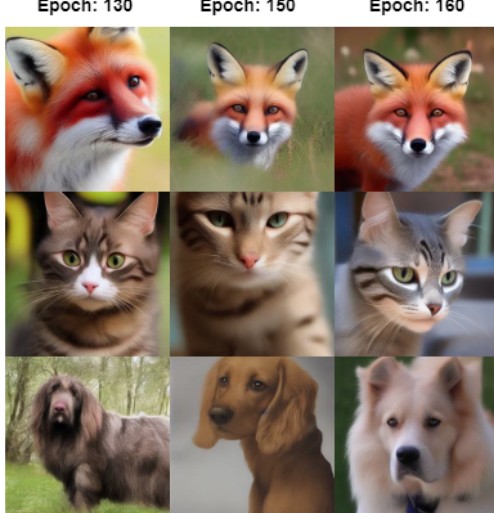

Figure 1: **Effect of Entropy Regularization on reward hacking.** This figure demonstrates the effectiveness of entropy regularization for mitigating reward hacking problems during Direct Preference Optimization (DPO) Xu et al. (2024). The **A** images are distorted whereas the **B** images are aesthetically better at the same epochs.

## 3   Background

**Diffusion Models:** Diffusion models are generative models that learn to represent the data distribution $p(\mathbf{x}_0)$ by gradually corrupting data with noise in a forward diffusion process, and subsequently learning to reverse this process. Denoising diffusion models parameterize the data distribution as $p_\theta(\mathbf{x}_0)$, where the reverse process is modeled as a Markov chain. This reverse process is discrete in time and follows the structure:

$$p_\theta(\mathbf{x}_{0:\mathbf{T}}) = p(x_T) \prod_{t=1}^{T} p_\theta(\mathbf{x}_{t-1} \mid \mathbf{x}_t) \tag{1}$$

where

$$p_\theta(x_{t-1} \mid x_t) = \mathcal{N}\left(x_{t-1}; \mu_\theta(x_t), \sigma_{t|t-1}^2 \frac{\sigma_{t-1}^2}{\sigma_t^2}\mathbf{I}\right)$$

$$p(x_T) = \mathcal{N}(x_T; 0, \mathbf{I})$$

The model is trained by maximizing the evidence lower bound (ELBO) on the data likelihood. The training objective for diffusion models is defined as:

$$L_{DM} = \mathbb{E}_{\mathbf{x}_0, \epsilon, t, \mathbf{x}_t}\left[\omega(\lambda_t)\|\epsilon - \epsilon_\theta(\mathbf{x}_t, t)\|_2^2\right] \tag{2}$$

Here, the noise $\epsilon$ is sampled from a normal distribution, i.e., $\epsilon \sim \mathcal{N}(\mu, \mathbf{I})$, and the time step $t$ is sampled uniformly from the range $t \sim \mathcal{U}(0, T)$ where $T$ is the total number of diffusion steps. The corrupted image $\mathbf{x}_t$ is sampled from a Gaussian distribution conditioned on the original image, $q(\mathbf{x}_t|\mathbf{x}_0) = \mathcal{N}(\mathbf{x_t}; \alpha_t\mathbf{x}_0, \sigma_t^2\mathbf{I})$,

where $\lambda_t = \alpha_t^2/\sigma_t^2$ represents the signal-to-noise ratio. The function $\omega(\lambda_t)$ is a predefined weighting function, often chosen as a constant for simplicity.

**Reward Modelling:** Reward modeling focuses on modeling human preferences for a generated sample $\mathbf{x}$ under a specific conditioning $c$. To achieve this a human-annotated dataset is usually used which contain triplets $(\mathbf{x_w}, \mathbf{x_l}, \mathbf{c})$, where $\mathbf{x_w}$ represents the preferred (winning) sample and $\mathbf{x_l}$ represents the less preferred (losing) sample given the conditioning $\mathbf{c}$.

For preference learning which is used in online iterative DPO, usually the Bradley-Terry (BT) model Bradley & Terry (1952) is used to describe the probability of $\mathbf{x_w}$ being preferred over $\mathbf{x_l}$ conditioned on $\mathbf{c}$, which is formalized as:

$$p_{\text{BT}}(\mathbf{x_w} \succ \mathbf{x_l}|\mathbf{c}) = \sigma\left(r(\mathbf{c}, \mathbf{x_w}) - r(\mathbf{c}, \mathbf{x_l})\right) \tag{3}$$

where $r(c, x)$ denotes the reward function and $\sigma(\cdot)$ is the sigmoid function. The reward function is parametrized as $r_\phi(c, x)$, where $\phi$ represents the model parameters, and we finally optimize the log-likelihood of Eq. 3 as the primary learning objective:

$$L_{\text{BT}}(\phi) = -\mathbb{E}_{\mathbf{c}, \mathbf{x_w}, \mathbf{x_l}}\left[\log \sigma\left(r_\phi(\mathbf{c}, \mathbf{x_w}) - r_\phi(\mathbf{c}, \mathbf{x_l})\right)\right] \tag{4}$$

**DPO:** Reinforcement Learning from Human Feedback (RLHF) aims to optimize the parameters $\theta$ of a conditional distribution $p_\theta(\mathbf{x}_0|\mathbf{c})$ over latents $\mathbf{x}_0$ conditioned on context $\mathbf{c}$. The objective of RLHF is commonly framed as maximizing the expected reward while minimizing divergence from a reference policy. Mathematically, the objective can be expressed as:

$$\max_{p_\theta} \mathbb{E}_{c \sim \mathcal{D}_c, x_0 \sim p_\theta(x_0|c)}\left[r(c, x_0)\right] - \beta \mathcal{D}_{\text{KL}}\left[p_\theta(x_0|c)\|p_{\text{ref}}(x_0|c)\right] \tag{5}$$

In Eq. 5, the first term encourages the policy to maximize the expected reward, while the second term, weighted by a hyperparameter $\beta$, introduces a regularization to keep the current policy close to a reference policy $p_{\text{ref}}$. In this formulation, $\beta$ controls the trade-off between exploration (reward maximization) and exploitation (closeness to the reference policy).

However, this formulation assumes that the reward model is ideal, which is often not the case in practice. Training and querying a reward model typically incurs high computational costs, making the approach challenging to scale. To address this issue, DPO Rafailov et al. (2024b) reformulates the problem to directly solve for the optimal policy $p_\theta^*$ in a closed form, defined as:

$$p_\theta^*(\mathbf{x}_0|c) = p_{\text{ref}}(\mathbf{x}_0|c) \exp\left(\frac{r(\mathbf{c}, \mathbf{x}_0)}{\beta}\right) / Z(c) \tag{6}$$

Here, $Z(c)$ is the partition function:

$$Z(c) = \sum_{x_0} p_{\text{ref}}(x_0|c) \exp\left(\frac{r(c, x_0)}{\beta}\right) \tag{7}$$

This leads to a rewritten form of the reward function:

$$r(c, x_0) = \beta \log \frac{p_\theta^*(x_0|c)}{p_{\text{ref}}(x_0|c)} + \beta \log Z(c) \tag{8}$$

Substituting this into the original objective in Eq. 8, the reward objective for DPO becomes:

$$L_{\text{DPO}}(\theta) = -\mathbb{E}_{c, x_0^w, x_0^l}\left[\log \sigma\left(\beta \log \frac{p_\theta(x_0^w|c)}{p_{\text{ref}}(x_0^w|c)} - \beta \log \frac{p_\theta(x_0^l|c)}{p_{\text{ref}}(x_0^l|c)}\right)\right] \tag{9}$$

This reparameterization allows DPO to bypass the need for reward function optimization and instead directly optimizes the conditional distribution $p_\theta(x_0|c)$. All DPO-based diffusion methods (Diffusion-DPO, D3PO & SPO) introduce approximation techniques to apply this approach to diffusion models.

**D3PO:** Instead of directly optimizing the reward model, Yang et al. (2024) formulate the generation process as a Markov Decision Process (MDP) and focus on learning a policy that maximizes the action-value function. The corresponding objective is given in Eq. 10.

$$\max_{\pi} \mathbb{E}_{s \sim d^{\pi}, a \sim \pi(\cdot|s)} \Big[ Q^*(s, a) \Big] - \beta \bigg( D_{\mathrm{KL}} \Big[ \pi(a|s) \| \pi_{\mathrm{ref}}(a|s) \Big] \bigg) \tag{10}$$

As we can see Eq. 10 is similar to Eq. 5 with $Q(s, a)$ inplace of $r(c, x_0)$. Consequently, we can adopt an optimization approach similar to DPO to derive their final loss function:

$$L_{\mathrm{D3PO}}(\theta) = -\mathbb{E}_{(s_k, \sigma_w, \sigma_t)} \left[ \log \rho \bigg( \beta \cdot \bigg( \log \frac{\pi_\theta(a_k^w|s_k^w)}{\pi_{\mathrm{ref}}(a_k^w|s_k^w)} - \log \frac{\pi_\theta(a_k^t|s_k^t)}{\pi_{\mathrm{ref}}(a_k^t|s_k^t)} \bigg) \bigg) \right] \tag{11}$$

**Diffsuion-DPO:** To address the memory limitations that arise when directly applying DPO to diffusion models Wallace et al. (2023) upper bound Eq. 9 using the Jensen-Shannon inequality. They approximate the reverse process $p_\theta(x_{1:T}|x_0)$ with the forward process $q(x_{1:T}|x_0)$ and leverage the fact that the forward process follows a Gaussian distribution to derive their final objective, as shown below:

$$L_{\mathrm{DiffusionDPO}}(\theta) = -\mathbb{E}_{t, \epsilon^w, \epsilon^l} \log \sigma \bigg( -\beta T \bigg[ \Big( \|\epsilon^w - \epsilon_\theta(x_t^w, t)\|^2 - \|\epsilon^l - \epsilon_\theta(x_t^l, t)\|^2 \Big)$$
$$- \Big( \|\epsilon^w - \epsilon_{\mathrm{ref}}(x_t^w, t)\|^2 - \|\epsilon^l - \epsilon_{\mathrm{ref}}(x_t^l, t)\|^2 \Big) \bigg] \bigg) \tag{12}$$

**SPO:** D3PO assumes that a preferred image is associated with a corresponding preferred diffusion chain, which may be a strong assumption. In contrast, SPO follows the same MDP formulation as D3PO but, instead of optimizing an action-value function, utilizes a per-step reward model capable of scoring noisy images and providing reliable preference signals at each step. This allows SPO to generate preferences iteratively without relying on D3PO's assumption. As a result, SPO can optimize Eq. 9 at each diffusion step, leading to a final objective that aligns with D3PO but is applied at every step of the diffusion process.

In this paper, we show that all these approaches are mathematically similar and we further introduce our self-entropy regularization in this unified framework.

## 4 Methodology

In this section, we prove the equivalence of D3PO, SPO, and Diffusion-DPO. We model the diffusion process as an MDP, we borrow the notations from Yang et al. (2024):

$$s_t \triangleq (c, t, x_{T-t})$$
$$P(s_{t+1} \mid s_t, a_t) \triangleq (\delta_c, \delta_{t+1}, \delta x_{T-1-t})$$
$$a_t \triangleq x_{T-1-t}$$
$$\pi(a_t \mid s_t) \triangleq p_\theta(x_{T-1-t} \mid c, t, x_{T-t})$$
$$r(s_t, a_t) \triangleq r\left( (c, t, x_{T-t}), x_{T-t-1} \right)$$

Where $T$ is the maximum number of denoising timesteps and $\delta_x$ represents a Dirac delta distribution. We want to optimize Eq. 9, we can re-write it as:

$$L_{\mathrm{DPO}}(\theta) = -\mathbb{E}_{c, x_0^w, x_0^l} \left[ \log \sigma \bigg( \beta \log \prod_i \frac{p_\theta(x_{t-1}^w|x_t^w)}{p_{\mathrm{ref}}(x_{t-1}^w|x_t^w)} - \beta \log \prod_i \frac{p_\theta(x_{t-1}^l|x_t^l)}{p_{\mathrm{ref}}(x_{t-1}^l|x_t^l)} \bigg) \right] \tag{13}$$

We assume that the number of genration timestep ($T$) is same for both winning and lossing sample, by using the property $\log x1 \cdot x2 = \log x1 + \log x2 \; \forall x1, x2 \in R^+$ we simplify 13. We don't write the conditioning on $c$ explicitly in Eq. 13 for conciseness:

$$L_{\text{DPO}}(\theta) = -\mathbb{E}_{c, x_0^w, x_0^l} \left[ \log \sigma \left( \beta T * \frac{1}{T} * \sum_{t=1}^{T} \log \frac{p_\theta(x_{t-1}^w | x_t^w)}{p_{\text{ref}}(x_{t-1}^w | x_t^w)} - \beta T * \frac{1}{T} * \sum_{t=0}^{T} \log \frac{p_\theta(x_{t-1}^l | x_t^l)}{p_{\text{ref}}(x_{t-1}^l | x_t^l)} \right) \right] \tag{14}$$

We can define $t$ as a random variable which is sampled from a uniform distribution with support from 0 to $T$. Using this and Jenson Shannon inequality we arrive at:

$$L_{\text{DPO}}(\theta) \le -\mathbb{E}_{c, x_0^w, x_0^l, t \sim \mathcal{U}(0, T)} \left[ \log \sigma \left( \beta T \log \frac{p_\theta(x_{t-1}^w | x_t^w)}{p_{\text{ref}}(x_{t-1}^w | x_t^w)} - \beta T \log \frac{p_\theta(x_{t-1}^l | x_t^l)}{p_{\text{ref}}(x_{t-1}^l | x_t^l)} \right) \right] \tag{15}$$

This objective is similar to Diffusion-DPO, if instead we consider optimization at each step then the objective translates to SPO and D3PO's objective. Now we introduce our self-entropy regularization into this framework. We start with D3PO's formulation:

$$\max_\pi \mathbb{E}_{s \sim d^\pi, a \sim \pi(\cdot|s)} \left[ Q^*(s, a) \right] - \beta \left( D_{\text{KL}} \left[ \pi(a|s) \| \pi_{\text{ref}}(a|s) \right] + \gamma \pi(a|s) \log \pi(a|s) \right)$$

$$\max_\pi \mathbb{E}_{s \sim d^\pi, a \sim \pi(\cdot|s)} \left[ Q^*(s, a) \right] - \beta \left( \pi(a|s) \log \frac{\pi(a|s)}{\pi_{\text{ref}}(a|s)} + \gamma \pi(a|s) \log \pi(a|s) \right)$$

$$\max_\pi \mathbb{E}_{s \sim d^\pi, a \sim \pi(\cdot|s)} \left[ Q^*(s, a) \right] - \beta \left( (\gamma + 1) \pi(a|s) \log \frac{\pi(a|s)}{\pi_{\text{ref}}(a|s)^{\frac{1}{\gamma+1}}} \right)$$

$$\max_\pi \mathbb{E}_{s \sim d^\pi, a \sim \pi(\cdot|s)} \left[ Q^*(s, a) \right] - \beta(\gamma + 1) \left( D_{\text{KL}} \left[ \pi(a|s) \| \pi_{\text{ref}}(a|s)^{\frac{1}{\gamma+1}} \right] \right) \tag{16}$$

We can see that the only effect of adding our self entropy term is on $\pi_{ref}$, particularly we can simplify Eq. 16 using the usual DPO derivation Rafailov et al. (2024b) to show that adding the self entropy is equivalent to raising $\pi_{ref}$ by $1 + \gamma$. Hence we can use the same procedure as D3PO to arrive at our update objective:

$$L(\theta) = -\mathbb{E}_{(s_k, \sigma_w, \sigma_t)} \left[ \log \rho \left( \beta(\gamma + 1) \cdot \left( \log \frac{\pi_\theta(a_k^w | s_k^w)}{\pi_{\text{ref}}(a_k^w | s_k^w)^{\frac{1}{\gamma+1}}} - \log \frac{\pi_\theta(a_k^t | s_k^t)}{\pi_{\text{ref}}(a_k^t | s_k^t)^{\frac{1}{\gamma+1}}} \right) \right) \right] \tag{17}$$

Since SPO follows the same formulation as D3PO Eq. 17 can also be applied to train SPO. From Eq. (17) we can see that for $\gamma > 0$ the final effect of adding the self-entropy term was to flatten out the reference distribution which in return forces the policy to be more explorative (appendix C.1), we can also write this in terms of the noise objective, similar to Diffusion-DPO:

$$L_w = (1 + \gamma) \left( \| \epsilon^w - \epsilon_\theta(x_t^w, t) \|^2 - \| \epsilon^l - \epsilon_\theta(x_t^l, t) \|^2 \right)$$

$$L_l = \left( \| \epsilon^w - \epsilon_{\text{ref}}(x_t^w, t) \|^2 - \| \epsilon^l - \epsilon_{\text{ref}}(x_t^l, t) \|^2 \right)$$

$$L_{\text{final}}(\theta) \le -\mathbb{E}_{t, \epsilon^w, \epsilon^l} \log \sigma \left( -\beta T (L_w - L_l) \right).$$

We could also start from the formulation of Diffusion-DPO to get an alternate version of our objective which is different only by a constant, we follow the same steps as of Diffusion-DPO to arrive at our final objective the exact derivation can be found in the appendix C.2:

$$L_w = \left( \left\| \epsilon^w - \epsilon_\theta(x_t^w, t) \right\|^2 - \left\| \epsilon^l - \epsilon_\theta(x_t^l, t) \right\|^2 \right)$$

$$L_l = \frac{1}{(1+\gamma)} \left( \left\| \epsilon^w - \epsilon_{\text{ref}}(x_t^w, t) \right\|^2 - \left\| \epsilon^l - \epsilon_{\text{ref}}(x_t^l, t) \right\|^2 \right)$$

$$L_{\text{final}}(\theta) \leq -\mathbb{E}_{t, \epsilon^w, \epsilon^l} \log \sigma \Big( -\beta T (L_w - L_l) \Big).$$

We can see that to a constant factor this is similar to one derived directly from Eq. 17. Intuitively, in this formulation, the loss associated with the reference model is reduced when $\gamma > 0$. Therefore the resulting diffusion model relies less on the reference model. This allows the trained model to explore the low probability space of the reference model and thus increases the diversity of the trained model.

# 5 Experiments

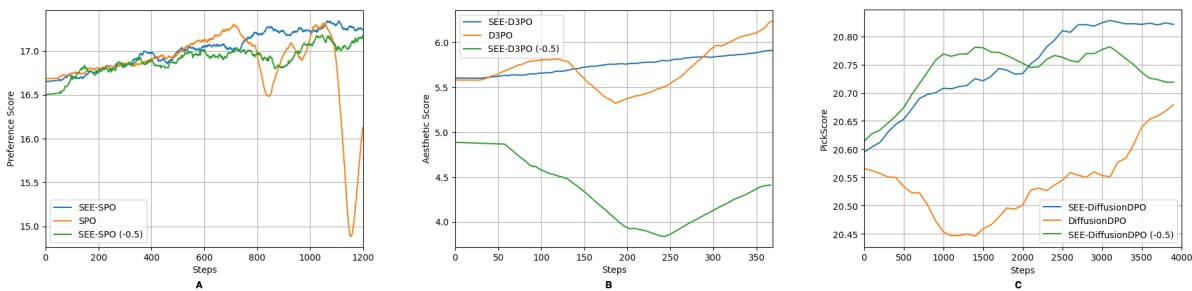

Figure 2: **Accuracy vs Steps** A: Step-Wise PickScore Liang et al. (2024), B: Aesthetic ScoreWang et al. (2022), C: PickScore Kirstain et al. (2023), demonstrate faster convergence, improved training stability and robustness to reward hacking.

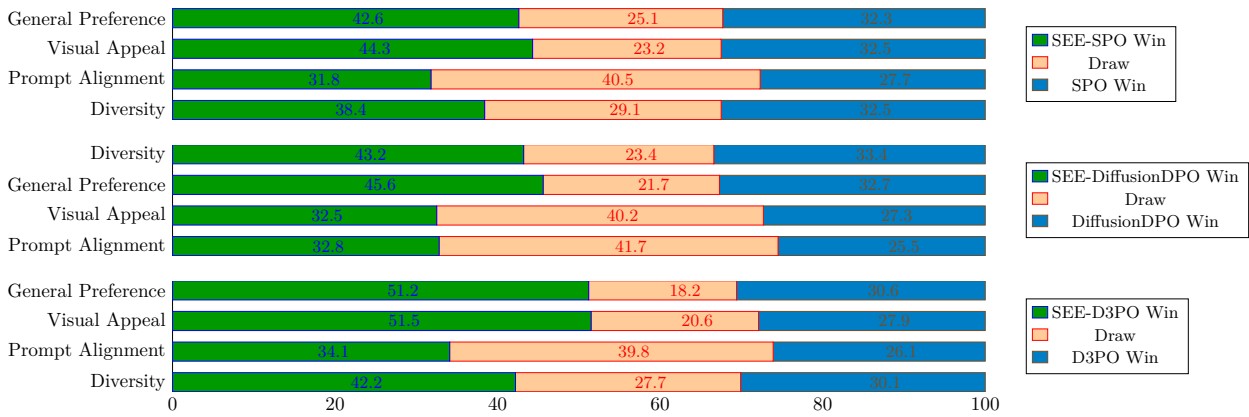

Figure 3: **User study.** We compare our best models with counterparts fine-tuned using the PickScore reward model Kirstain et al. (2023). Following SPO Liang et al. (2024), we randomly select 300 prompts from PartiPrompts Yu et al. (2022) and HPS Wu et al. (2023) in a 1:2 ratio. Participants are shown a reference image alongside three generated images from the same prompt but with different initial latents. They evaluate image quality and diversity relative to the reference to estimate the diversity metric. For other metrics, users assess single images.

**Experimental Setting:** For a fair comparison, we used the official implementations of D3PO, Diffusion-DPO, and SPO with their default parameters. We trained these models using our proposed regularized loss

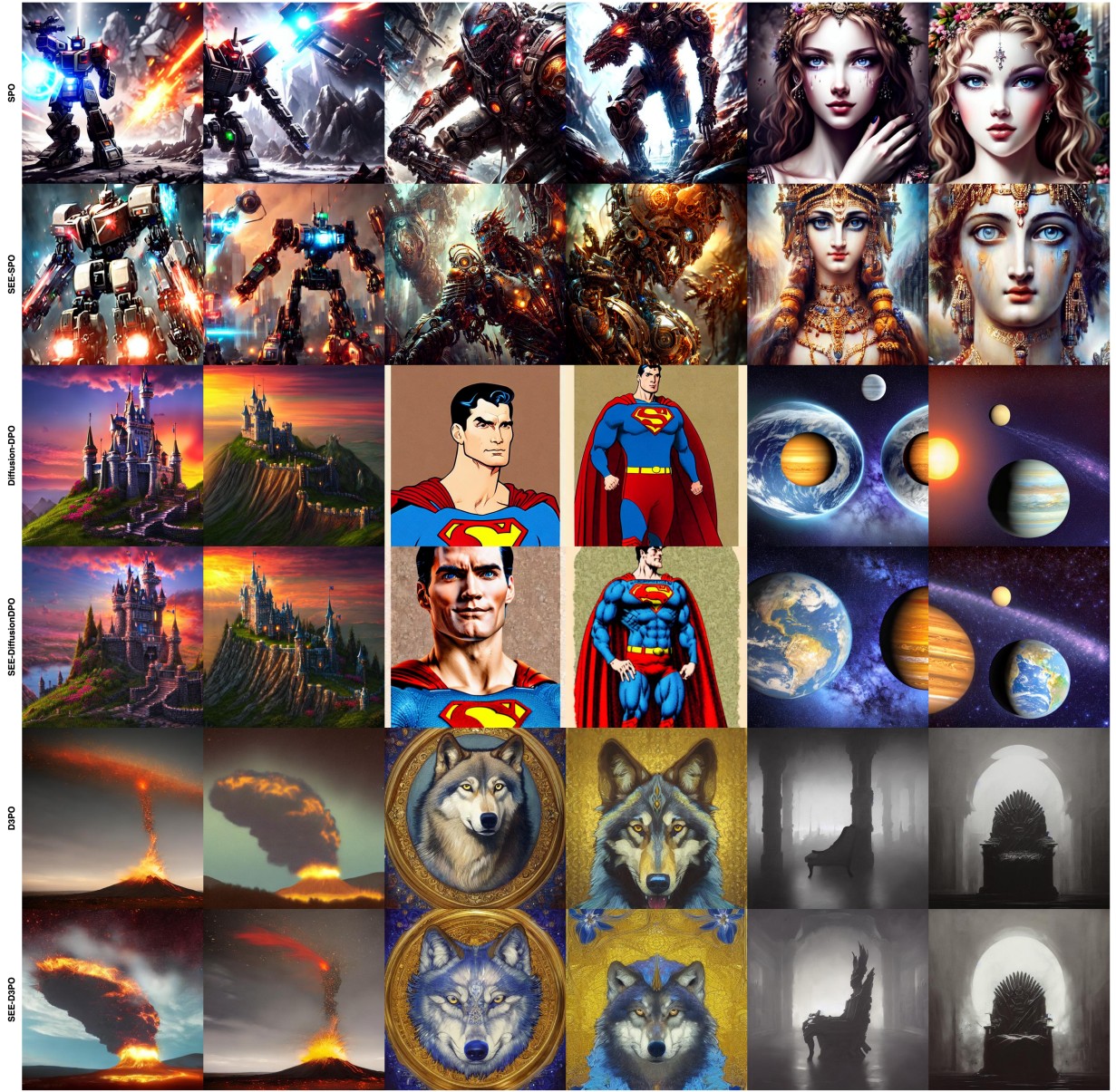

Figure 4: **Enhancing Generation Diversity.** Our method demonstrates superior image quality, producing clearer and more detailed results across different latent variables. In contrast, existing approaches show significant degradation in image quality, with blurring and distortions becoming evident as latent variations increase.

| Method | PickScore ↑ | HPS ↑ | Aesthetic Score ↑ | BLIP ↑ | ImageReward ↑ | CLIP ↑ |
|---|---|---|---|---|---|---|
| D3PO | 20.306 | 0.227 | 5.050 | 0.470 | -0.209 | 0.261 |
| Diffusion-DPO | 20.791 | 0.262 | 4.711 | 0.486 | 0.139 | 0.263 |
| SPO | 20.919 | 0.266 | 4.702 | 0.472 | 0.191 | 0.249 |
| SEE-D3PO | 20.590 | 0.247 | 4.851 | 0.491 | -0.003 | 0.271 |
| SEE-DiffusionDPO | 21.014 | 0.273 | 4.721 | 0.497 | 0.333 | 0.266 |
| SEE-SPO | 21.295 | 0.287 | 4.679 | 0.510 | 0.585 | 0.275 |

Table 1: **Quality Metrics.** All numbers are reported on the validation-unique split of Pick-a-Pic V1 dataset. We evaluate on CLIP Score Hessel et al. (2022), BLIP Score Li et al. (2022), Aesthetic Score Wang et al. (2022), PickScore Kirstain et al. (2023), HPSv2 Score Wu et al. (2023)

| Method | RMSE ↑ | PSNR ↓ | SSIM ↓ | E1 ↑ | E2 ↑ |
|---|---|---|---|---|---|
| StableDiffusion-1.5 | 0.3745 | 8.6943 | 0.2100 | 7.4627 | 7.4620 |
| Diffusion-DPO | 0.3736 | 8.7047 | 0.2406 | 7.4954 | 7.4945 |
| SEE-DiffusionDPO | 0.4299 | 8.1743 | 0.2056 | 7.6517 | 7.6587 |
| D3PO | 0.2455 | 12.5599 | 0.4190 | 7.0853 | 7.0911 |
| SEE-D3PO | 0.3052 | 10.5240 | 0.3183 | 7.3650 | 7.3692 |
| SPO | 0.4446 | 7.1218 | 0.1666 | 7.5674 | 7.5635 |
| SEE-SPO | 0.4383 | 7.0818 | 0.1575 | 7.7120 | 7.7101 |

Table 2: **Diversity Metrics.** All values are reported on the validation-unique split of the Pick-a-Pic-V1 dataset Kirstain et al. (2023). For each prompt, we first sample a base image and then sample 10 different images to get all the diversity metrics. Evaluation metrics include RMSE, PSNR, SSIM, and entropy energy scores (E1 and E2) Sun et al. (2024).

function, as described in the Methodology section. When applying our method, we treated only $\gamma$ and $\beta$ as hyperparameters while keeping all other settings at their default values to ensure a fair evaluation. During inference, we set the guidance scale to 7.5 for consistency across models.

For training, we used 4,000 prompts from the Pick-a-Pic-V1 dataset Kirstain et al. (2023) for SPO and D3PO, following the dataset provided in SPO's official GitHub repository. For Diffusion-DPO, we used 800,000 prompts from the same dataset. Each model was trained using the same setup and data splits as specified in their original implementations. To establish a fair basis for comparison, we selected Stable-Diffusion v1.5 as the benchmark model.

Since our training approach is online, we relied on automatic preference generators. Specifically, we used the time-conditional PickScore Liang et al. (2024) as a proxy for human preference in SPO, human annotations from the Pick-a-Pic-V1 dataset for Diffusion-DPO, and PickScore Kirstain et al. (2023) for D3PO. Notably, we did not use online training for Diffusion-DPO to remain consistent with its original implementation.

**Evaluation:** To assess the quality of generated images, we utilize automatic evaluation metrics, including PickScore Kirstain et al. (2023), HPS Wu et al. (2023), Aesthetic Score Wang et al. (2022), CLIP Hessel et al. (2022), BLIP Li et al. (2022), and ImageReward Xu et al. (2023). We report results on the `validation_unique` split of the Pick-a-Pic V1 dataset, which contains 500 prompts, as shown in Tables 1, 2, 4 and 5.

For a comprehensive assessment of generation diversity, we employ five additional metrics. Root Mean Square Error (RMSE), Peak Signal-to-Noise Ratio (PSNR), and Structural Similarity Index Measure (SSIM) are commonly used to evaluate image similarity; here, we leverage them to quantify diversity in generated images. Additionally, we compute image entropy, including both 1D and 2D entropy, to measure the information content within images. Higher entropy values indicate greater diversity and richness in image content.

To evaluate diversity, we generate 10 images per prompt from the `validation_unique` set and compute the aforementioned diversity metrics Sun et al. (2024). We do not rely on embedding-based similarity, as images generated from the same prompt, despite visual differences, often cluster closely in embedding space. An effectively aligned model should achieve high reward scores while producing diverse outputs. Our results

demonstrate that the regularized versions of D3PO, SPO, and Diffusion-DPO achieve higher mean rewards across multiple metrics while improving diversity scores, highlighting enhanced performance in both diversity and image quality compared to baseline methods.

Additionally, we conduct a user study similar to Liang et al. (2024). We recruit 10 participants to evaluate images generated by different models based on 300 prompts sampled from PartiPrompts and HPSv2 in a 1:2 ratio. Participants assess four aspects: *Prompt Alignment*, *Visual Appeal*, *General Preference*, and *Diversity*. To evaluate diversity, we provide each participant with a reference image generated by a model alongside three additional images generated from different latent codes. Participants rate both their general preference and the diversity of the additional images based on their differences from the reference image.

| Model | $\gamma$ | $\beta$ |
|---|---|---|
| D3PO | 5 | 0.01 |
| Diffusion-DPO | 3 | 4 |
| SPO | 3 | 0.1 |

Table 3: **Hyperparameter values**: All the other hyperparameters values were fixed to the original implementation

### 5.1 Results

Our models, trained using the proposed objective, outperform or are comparable to baseline methods across all quality metrics, as shown in Table 2. Specifically, SEE-D3PO improves PickScore by 1.41%, HPS by 8.8%, and ImageReward by 98.5%. SEE-DiffusionDPO yields improvements of 1.08% in PickScore, 4.2% in HPS, and 139.5% in ImageReward. Lastly, SEE-SPO shows gains of 1.8% in SPO, 7.9% in HPS, and 206.3% in ImageReward.

In terms of image diversity metrics, all our models outperform their respective counterparts across all six metrics. This trend is also reflected in the user study [Fig: 3], where our methods achieve at least a 6% improvement in generation diversity. Furthermore, our approach enhances all other evaluation aspects, though we observe minimal gains in the Prompt Alignment task. Example samples are provided in the appendix and Fig: 4.

### 5.2 Ablations

In our experiments, we explored the effect of different values of $\gamma$ on both training stability and overall model performance. As shown in Fig: 2, when $\gamma = 0$, representing the baseline or vanilla model, we observe an initial steep drop in reward values, followed by a sudden and rapid increase during training phases of all the models SPO and D3PO training phases. Examining the generated images in this region, we note a gradual decline in quality corresponding to the initial drop in scores, followed by further degradation but the reward values continue to rise. This pattern signals that the model begins to overfit prematurely, leading to the generation of repetitive and noisy images—a phenomenon we define as reward hacking. In this scenario, the model exploits reward dynamics by producing images that maximize the reward function but lead to distorted images as shown in Fig: 1. Through our experimentation, we observed two distinct types of overfitting. In the first, seen in D3PO, the reward values continue to increase despite a progressive decline in image quality, indicating that the model is optimizing for reward at the expense of meaningful outputs. In the second, observed in SPO, the reward model assigns consistently low scores, and the model fails to improve beyond this point, becoming stuck in a suboptimal state. Notably, this trend is absent in the Diffusion-DPO setup, which utilizes the extensive Pick-a-Pic-V1 dataset containing over 800,000 images, likely providing more comprehensive training data that mitigates early overfitting. For diffusion DPO we see that the model with $\gamma = -0.5$ initially increases in accuracy and then decreases. Across all configurations, our model achieves higher scores faster, with more consistent stability in training and fewer abrupt changes in reward values.

As training progresses, our model not only stabilizes but also consistently reaches a higher reward range. We attribute this improvement to the implicit flattening of the reference distribution that higher $\gamma$ values

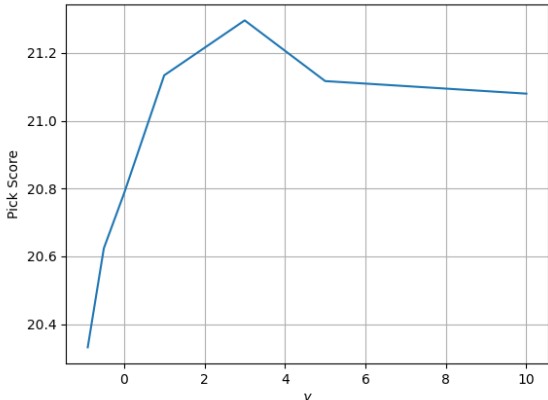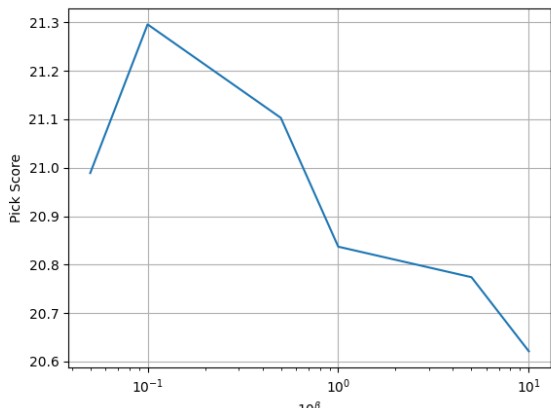

Figure 5: **Ablation Study for $\gamma$ & $\beta$**: This study investigates the impact of $\gamma$ and $\beta$ on image quality, specifically using PickScore as the metric. In the first ablation, we examine the effect of varying $\gamma$ while keeping all other parameters fixed, with $\beta$ set to 0.1. In the second ablation, we explore the influence of different values of $\beta$ for $\gamma = 3$ settings.

introduce. This flattening effect broadens the policy distribution, enabling more exploratory behavior and decreasing the likelihood of overfitting to repetitive patterns.

Further, we observe empirically poorer results compared to the base model when $\gamma = -0.5$, as shown in Fig: 3. D3PO demonstrates early overfitting tendencies, resulting in sharp reward increases followed by rapid declines. Meanwhile, Diffusion-DPO consistently underperforms relative to other models, as the negative $\gamma$ value restricts exploratory actions, yielding repetitive outputs that fail to capture meaningful image variations. In the SPO pipeline, overfitting occurs somewhat later than in the vanilla model. This difference can be attributed to SPO's update policy, which only updates the model when it generates images with reward differences that exceed a predetermined threshold. With a negative $\gamma$, the generation distribution narrows, limiting the diversity of generated images and, consequently, the frequency of model updates. Further evidence of these observations can be seen by examining the rewards vs. steps graph for SPO $\gamma = -0.5$, where the reward values remain nearly constant throughout most of the training duration before ultimately showing signs of overfitting. This steady reward trend reinforces the observation that the model fails to explore diverse solutions, hindering its ability to generalize and adapt, particularly under negative $\gamma$ settings.

The optimal values for $\beta$ and $\gamma$ are shown in Table 3. Notably, the optimal $\beta$ for Diffusion-DPO is two orders of magnitude larger than that of D3PO, which is expected since the $\beta$ for Diffusion-DPO is multiplied by $T$, a term of order 2. In the case of SPO, we update at each timestep, which is of order 1, so we also expect $\beta$ to increase by one order to prevent significant deviation from the reference distribution.

We also conducted a deeper ablation study using SPO, exploring various values of $\beta$ and $\gamma$. In Fig: 5, the left image shows the results of keeping $\beta$ fixed at 0.1 while varying $\gamma$. We observe that for $\gamma > 1$, all experiments achieve high PickScore, whereas for $\gamma < 0$, the performance significantly drops. As noted earlier, $\gamma < 0$ leads to reduced diversity and poor training, with results comparable to the vanilla Stable-Diffusion 1.5 model. In the right image, we fix $\gamma = 3$ and vary $\beta$ while keeping other hyperparameters constant. Initially, for low values of $\beta$, the performance is high, but it declines steeply as $\beta$ increases. This decline may be due to the model's increasing reliance on the initial distribution as the regularization strength $\beta$ rises, limiting exploration and slowing training, which results in lower scores.

# 6 Conclusion

In this paper, we introduced Self-Entropy Enhanced Direct Preference Optimization (SEE-DPO) for fine-tuning diffusion models. We showed existing approaches that optimize Eq. 5 often suffer from overfitting and distribution shift issues. To address these challenges, we introduced a self-entropy regularization term into the objective function, transforming it into the final optimization objective outlined in Eq. 17. Our empirical results demonstrate that the proposed SEE-DPO, SEE-DiffusionDPO, and SEE-DPO method significantly outperforms their respective counterparts, effectively mitigating the reward hacking problem commonly associated with RLHF approaches and resulting in robust training. Additionally, our method ensures diversity of image generation across the latent space, highlighting its robustness and effectiveness in maintaining high-quality outputs.

Our approach was straightforward to implement within the current framework, specifically DPO with KL-regularized reward optimization. However, extending our method to incorporate other forms of regularization, such as Hellinger distance, may introduce additional mathematical complexity. We leave the exploration of alternative regularizations and their impact on performance to future work. Furthermore, although our diversity metrics (RMSE and PSNR) showed minimal improvement with SPO, the user study reported a higher diversity. This discrepancy highlights the need for a diversity metric that better aligns with human judgment, which we plan to investigate in future work.

# 7 Impact Statement

Generative models for media, such as image generation, present both significant opportunities and challenges. On one hand, these models enable a wide range of creative applications and, with advancements that reduce training and inference costs, have the potential to make this technology more accessible and encourage broader exploration. On the other hand, they also pose risks by making it easier to create and disseminate manipulated data, which can contribute to the spread of misinformation and harmful content, including the prevalent issue of deepfakes. Additionally, generative models may inadvertently expose sensitive or personal information from their training data, raising concerns about privacy, particularly when such data is collected without consent. The potential for this issue in diffusion models, specifically in image generation, remains an area requiring further investigation. Moreover, deep learning systems, including diffusion models, can perpetuate or amplify biases inherent in their training data. Our methods explores increasing diversity of generation which could positively impact the bias-corretion problem. For a more detailed discussion of the ethical considerations of deep generative models we refer readers to Hagendorff (2024).

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

# A    Appendix

This supplementary document provides additional details to complement the main paper:

- Additional experiment results

- Additional qualitative results

- Proofs

# B    Additional Experiment Results

In this section, we present additional evidence of the statistical significance of our results, as shown in Tables 4 and 5. All experiments were conducted over five randomly chosen seeds to ensure robustness. Furthermore, for Fig: 6, we generated five images per prompt using different latent variables while keeping the prompts consistent.

Our model consistently outperforms its counterparts, producing images with significantly finer details, thereby demonstrating its robustness and effectiveness in generating high-quality outputs.

Among the evaluated visual metrics, we observe that SSIM consistently aligns with the visual improvements, we attribute this to the fact that it effectively measures fine-grained structural similarities. In contrast, RMSE and PSNR, being coarse-grained metrics, failed to account for the nuanced details in the generated images. These results highlight the importance of using metrics like SSIM for a more comprehensive evaluation of fine-grained image generation tasks.

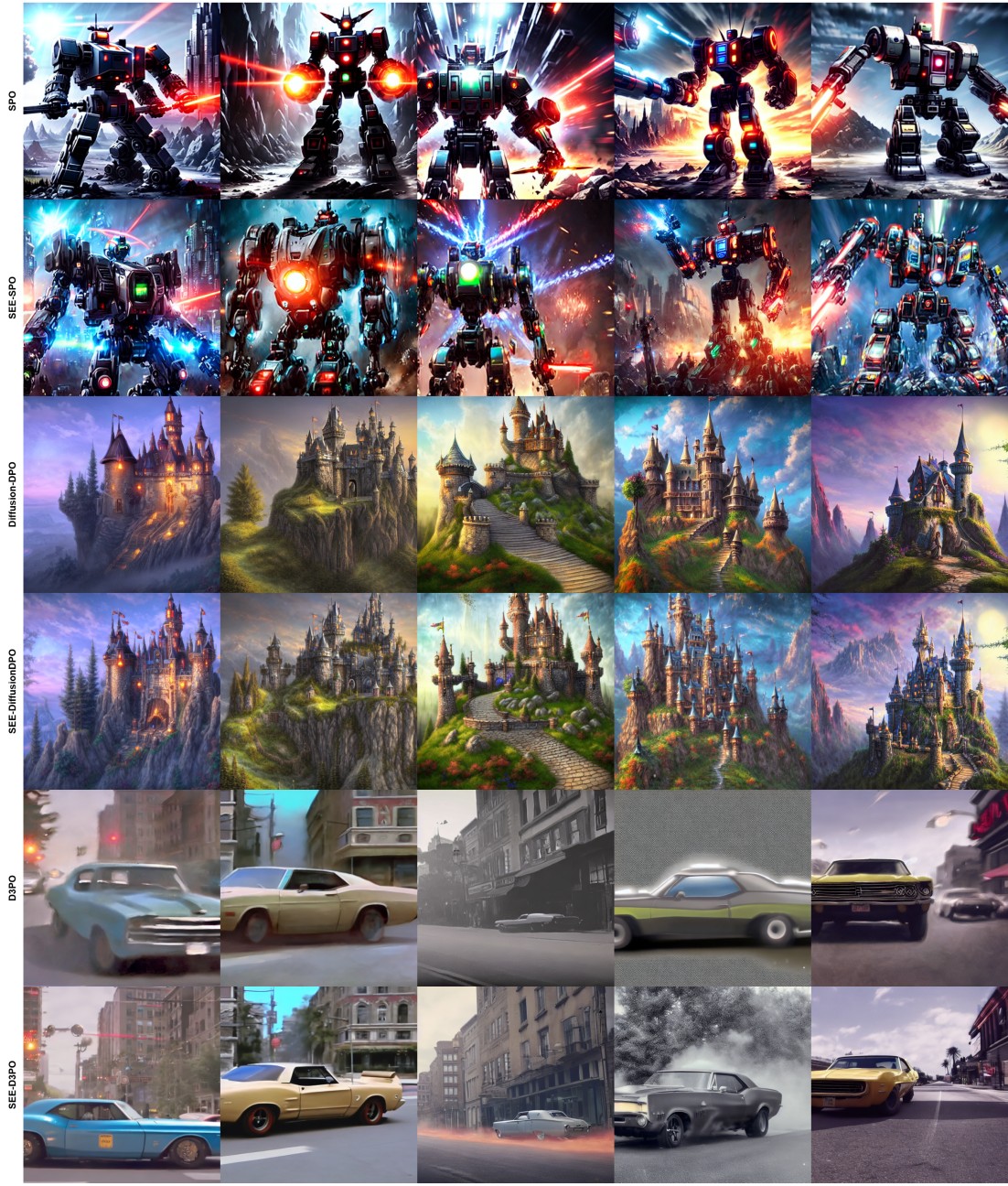

Figure 6: **Images are generated using the same prompts across 5 different latents, demonstrating that our models consistently outperform their counterparts across all cases.**

| Method | PickScore ↑ | HPS ↑ | Aesthetic Score ↑ | BLIP ↑ | ImageReward ↑ | CLIP ↑ |
|---|---|---|---|---|---|---|
| D3PO | $20.306 \pm 0.016$ | $0.227 \pm 0.001$ | $5.050 \pm 0.005$ | $0.470 \pm 0.001$ | $-0.209 \pm 0.038$ | $0.261 \pm 0.002$ |
| Diffusion-DPO | $20.791 \pm 0.021$ | $0.262 \pm 0.001$ | $4.711 \pm 0.010$ | $0.486 \pm 0.002$ | $0.139 \pm 0.034$ | $0.263 \pm 0.001$ |
| SPO | $20.919 \pm 0.024$ | $0.266 \pm 0.002$ | $4.702 \pm 0.019$ | $0.472 \pm 0.002$ | $0.191 \pm 0.028$ | $0.249 \pm 0.001$ |
| SEE-D3PO | $20.590 \pm 0.017$ | $0.247 \pm 0.001$ | $4.851 \pm 0.016$ | $0.491 \pm 0.001$ | $-0.003 \pm 0.018$ | $0.271 \pm 0.002$ |
| SEE-DiffusionDPO | $21.014 \pm 0.032$ | $0.273 \pm 0.003$ | $4.721 \pm 0.015$ | $0.497 \pm 0.002$ | $0.333 \pm 0.052$ | $0.266 \pm 0.001$ |
| SEE-SPO | $21.295 \pm 0.041$ | $0.283 \pm 0.001$ | $4.679 \pm 0.010$ | $0.510 \pm 0.001$ | $0.585 \pm 0.028$ | $0.275 \pm 0.001$ |

Table 4: **Results are averaged over 5 random seeds, with mean and standard deviation reported.** All numbers are reported on the validation-unique split of Pick-a-Pic V1 dataset. We evaluate on CLIP Score, BLIP Score , Aesthetic Score , PickScore , HPSv2 Score

| Method | RMSE ↑ | PSNR ↓ | SSIM ↓ | E1 ↑ | E2 ↑ |
|---|---|---|---|---|---|
| D3PO | $0.2455 \pm 0.0006$ | $12.5599 \pm 0.0248$ | $0.4190 \pm 0.0014$ | $7.0853 \pm 0.0075$ | $7.0911 \pm 0.0074$ |
| Diffusion-DPO | $0.3736 \pm 0.0017$ | $8.7047 \pm 0.0401$ | $0.2406 \pm 0.0013$ | $7.4954 \pm 0.0071$ | $7.4945 \pm 0.0070$ |
| SPO | $0.4446 \pm 0.0008$ | $7.1218 \pm 0.0154$ | $0.1666 \pm 0.0015$ | $7.5674 \pm 0.0118$ | $7.5635 \pm 0.0118$ |
| SEE-DiffusionDPO | $0.4299 \pm 0.0005$ | $8.1743 \pm 0.0095$ | $0.2056 \pm 0.0025$ | $7.6517 \pm 0.0059$ | $7.6587 \pm 0.0060$ |
| SEE-D3PO | $0.3052 \pm 0.0003$ | $10.5240 \pm 0.0100$ | $0.3183 \pm 0.0021$ | $7.3650 \pm 0.0123$ | $7.3692 \pm 0.0124$ |
| SEE-SPO | $0.4383 \pm 0.0021$ | $7.0818 \pm 0.0351$ | $0.1575 \pm 0.0018$ | $7.7120 \pm 0.0072$ | $7.7101 \pm 0.0074$ |

Table 5: **Results are averaged over 5 random seeds, with mean and standard deviation reported.** All values are reported on the validation-unique split of the Pick-a-Pic-V1 dataset. For each prompt, we first sample a base image and then sample 10 different images to get all the diversity metrics. Evaluation metrics include RMSE, PSNR, SSIM, and entropy energy scores (E1 and E2).

## C  Proofs

### C.1  Effects of $\gamma$

The parameter $\gamma$ serves to flatten the reference distribution. As seen in Eq. 16, the objective includes a KL divergence term that penalizes deviation from the reference policy, which in turn can restrict the training policy from exploring potentially higher-reward regions. By flattening the reference distribution via a higher $\gamma$, the training policy is allowed greater flexibility to explore a wider range of actions. This effect is illustrated in Fig. 7, where the high-reward region is initially assigned very low probability under the standard reference policy, making it difficult for the training policy to reach it—resulting in slower learning. In contrast, with a larger $\gamma$, the reference distribution becomes more uniform, assigning greater probability mass to the high-reward region. This facilitates more effective exploration and accelerates training.

### C.2  Diffusion-DPO equivalence

We could also start from the formulation of Diffusion-DPO to get an alternate version of our objective which is different only by a constant, we follow the same steps as of Diffusion-DPO to arrive at our final objective:

$$\max_{p_\theta} \mathbb{E}_{c \sim \mathcal{D}_c, x_{0:T} \sim p_\theta(x_{0:T}|c)}[r(c, x_0) - \beta \mathbb{D}_{\mathrm{KL}}[p_\theta(x_{0:T} \mid c) \| p_{\mathrm{ref}}(x_{0:T} \mid c)] - \beta \gamma p_\theta(x_{0:T} \mid c) \log p_\theta(x_{0:T} \mid c) \quad (18)$$

Further let $r(c, x_0) = \mathbb{E}_{p_\theta(x_{1:T}|x_0,c)}[R(c, x_{0:T})]$, plugging this to Eq. 18, upper-bounding KL-divergence and using jensen's inequality we get our objective:

$$L_w = \omega(\lambda_t) \left( \|\epsilon^w - \epsilon_\theta(x_t^w, t)\|^2 - \|\epsilon^l - \epsilon_\theta(x_t^l, t)\|^2 \right)$$

$$L_l = \omega(\lambda_t(1+\gamma)) \left( \|\epsilon^w - \epsilon_{\mathrm{ref}}(x_t^w, t)\|^2 - \|\epsilon^l - \epsilon_{\mathrm{ref}}(x_t^l, t)\|^2 \right)$$

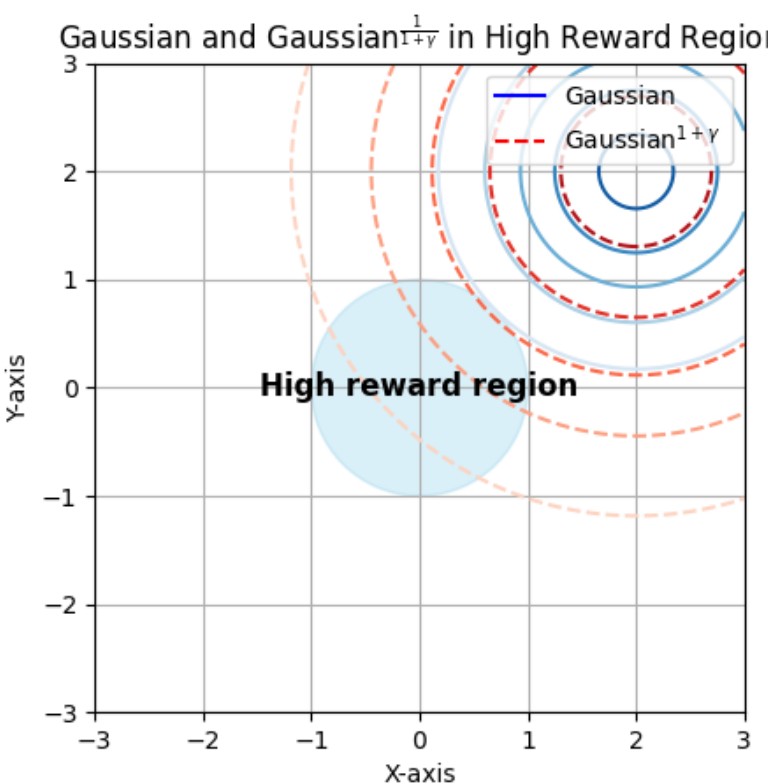

Figure 7: **A toy example illustrating how incorporating $\gamma$ encourages greater exploration, potentially guiding the model toward regions with higher rewards.**

$$L_{\text{final}}(\theta) \leq -\mathbb{E}_{t,\epsilon^w,\epsilon^l} \log \sigma \left( -\beta T \left( L_w - L_l \right) \right)$$

Further $\omega(t) \propto 1/\sigma_t^2$, here $\sigma_t^2$ refers to the variance at timestep $t$. In most work Wallace et al. (2023); Song et al. (2022); Ho et al. (2020) $\omega(t)$ is taken as a constant. Combining both we write our final objective as:

$$L_w = \left( \left\| \epsilon^w - \epsilon_\theta(x_t^w, t) \right\|^2 - \left\| \epsilon^l - \epsilon_\theta(x_t^l, t) \right\|^2 \right)$$

$$L_l = \frac{1}{(1+\gamma)} \left( \left\| \epsilon^w - \epsilon_{\text{ref}}(x_t^w, t) \right\|^2 - \left\| \epsilon^l - \epsilon_{\text{ref}}(x_t^l, t) \right\|^2 \right)$$

$$L_{\text{final}}(\theta) \leq -\mathbb{E}_{t,\epsilon^w,\epsilon^l} \log \sigma \left( -\beta T(L_w - L_l) \right).$$

