# OpenReview forum: "SEE-DPO: Self Entropy Enhanced Direct Preference Optimization"
_TMLR — Accepted by TMLR_

### Review · Reviewer_NzNp · 2025-02-26

**Summary Of Contributions:**

The paper introduces Self-Entropy DPO (SEE-DPO), a method designed to mitigate the reward hacking phenomenon during DPO for diffusion models by incorporating a self-entropy regularization term. They prove the equivalence of three DPO variants (D3PO, SPO, and Diffusion-DPO) and demonstrate that the proposed regularization can be applied to all of them. Experimental results validate the effectiveness of the regularization across these three variants, using Stable Diffusion v1.5 as the benchmark model.

**Audience:**

Yes

**Claims And Evidence:**

Yes

**Requested Changes:**

I recommend that the authors carefully proofread the paper and ensure formatting and text consistency. Some issues that need to be addressed include:

1. Table 5 in Appendix B: the rows are out of order, could the authors please arrange them properly, matching the format of Table 4 for easy comparison.
2. Page 6, first line: There should be a space before "Consequently, we …".
3. Page 6, Equation (11): It is inappropriate to state "derive our final loss function" since this function was originally proposed by D3PO. Please clarify this.
4. Inconsistent formatting in captions: Captions for Table 1, Table 2, Figure 4, and Figure 6 are bolded, whereas other figures and tables are not. Please ensure consistency.
5. I think Figure 4 is not referenced anywhere in the text, please consider include a reference and explanation (maybe in first paragraph of page 11).
6. Inconsistent figure notations: Some figures are labeled as "Figure X" while others use "Fig: X" (e.g., "Figure 5" vs. "Fig: 3"). Please standardize the notation.

**Strengths And Weaknesses:**

**Strength:**
   * The proposed regularization is general and straightforward.
   * It can be applied to multiple DPO variants, including D3PO, SPO, and Diffusion-DPO, and its effectiveness is demonstrated.

**Weakness**
   * While effective, entropy regularization is not a novel concept in RL.
   * The experiments are limited to Stable Diffusion v1.5.
   * Figure 2 is difficult to interpret based on the description on page 11. The text mentions an initial steep drop in reward values during both SPO and D3PO training phases, but the explanation is unclear. Could the authors please elaborate on how to properly interpret these plots?
   * Please see the requested changes.

---

> ### Author Response · Authors · 2025-03-05
> **Rebuttal**
>
> Thank you very much for your feedback and time spent reviewing our paper. We have carefully addressed all the formatting and consistency issues and have submitted a revised version with the necessary corrections.
> Beside that we have also edited the Fig: 4 explanation to better explain the figure.
>
>
> **Figure 4 Explanation**: In Figure 4, we observe an initial decline in reward values for both SPO and D3PO. Following this decline, both methods show an increase in reward values. However, in the case of D3PO, the reward continues to increase, while the image quality progressively deteriorates. This suggests that the model is exploiting the reward function by generating out-of-distribution samples that maximize the reward while deviating from meaningful image generation.
>
> For SPO, a similar trend is observed initially, where the reward increases after the initial decline. However, unlike D3PO, the reward then drops to a significantly lower level and never recovers. This represents a more severe form of model collapse, where the model consistently generates low-reward images and remains stuck in this suboptimal state, unable to recover.
>
> We report results on Stable Diffusion-v1.5 due to hardware constraints and to also give a fair comparison for all methods, as D3PO only uses Stable Diffusion-v1.5.

---

### Review · Reviewer_1Kva · 2025-03-25

**Summary Of Contributions:**

This paper presents a self-entropy regularization mechanism to overcome the problems of overfitting and reward hacking for Direct Preference Optimization (DPO) methods. The authors present some results to show that their SEE-DPO method can improve DPO training with better network stability, result quality and image diversity.

**Audience:**

Yes

**Claims And Evidence:**

Yes

**Requested Changes:**

I am not an expert in this area. To be honest, I have no idea of what requested changes the authors should have. So I will count on other reviewers' comments.

**Strengths And Weaknesses:**

**Strengths**:

1. The problem to address is important and will have a lot of interesting applications.
2. The authors provide some theoretical analysis and formulate their method as a self-entropy  regularization trick.
3. The paper is generally well written. I can follow most of the things. And the related work part is comprehensive which details the literature contexts for me.

**Weakness**:
1. In Table 1, the SEE based methods have similar metric results to other baselines, like Diffusion-DPO. Could you please provide some explanation?
2. Although I do not want to be picky, I cannot perceive notable visual improvements over baseline in Figure 3. For instance, for the last example, the fox generated by D3PO even presents more details than the proposed method. And the results of both methods are quite close in the first row.
3. The visual results are quite limited. I can only see comparisons in Figure 3 and Figure 6.

---

> ### Author Response · Authors · 2025-04-24
> **Official Comment by Authors**
>
> Thank you very much for your insightful feedback and for taking the time to review our paper.
>
> Our primary objective with this work is to enhance the diversity of generated outputs. As shown in Table 2, the SEE-based methods demonstrate a notable improvement in diversity metrics, achieving approximately a 15% improvement in RMSE, 14.5% in SSIM, 6.1% in PSNR, and 2.6% in entropy, when compared to baseline approaches such as Diffusion-DPO. Importantly, these gains in diversity are achieved while also improving quality metrics: for DiffusionDPO, we observe an increase of 1.41% in PickScore, 8.8% in HPS, and a significant 139.5% in ImageReward. While the gain in Aesthetic Reward is comparatively modest, this metric primarily captures image quality rather than prompt-image alignment. Moreover, we found it to be somewhat unstable as an evaluation signal, though we chose to report it due to its common usage in the literature.
>
> Regarding prompt-image alignment metrics such as CLIPScore and BLIPScore, we note that these remain relatively unchanged—a trend consistent across other related approaches like DDPO, DPOK, and D3PO. This suggests that while our method emphasizes enhancing diversity, it maintains comparable alignment quality with existing baselines.
>
> Additionally, we observe that PickScore can be somewhat insensitive in its scaling: for instance, although there is a clear visual and qualitative improvement from DiffusionDPO to SPO (especially with SDXL), the PickScore only marginally increases from 22.64 to 23.06. This highlights certain limitations in current evaluation metrics, which we believe warrants further discussion in future work.

---

### Review · Reviewer_ruq3 · 2025-04-17

**Summary Of Contributions:**

This paper provides a simple yet effective contribution to the field of generative model fine-tuning. It tackles an important problem – the tendency of RLHF methods to collapse model diversity and overfit to the reward – specifically in the context of diffusion models. The proposed solution, incorporating a self-entropy term into the preference optimization objective, is well-motivated. Empirical results indicate that this solution improves both quantitative metrics and qualitative outcomes.

**Audience:**

Yes

**Broader Impact Concerns:**

The paper raises several ethical considerations that should be added in an explicit Broader Impact Statement section, such as misuse for synthetic misinformation and deepfakes, amplification of bias and stereotypes, and copyright issues.
We recommend the authors include a dedicated Broader Impact section that acknowledges these potential harms and outlines mitigation strategies.

**Claims And Evidence:**

Yes

**Requested Changes:**

- Improve theoretical clarity: While the core idea comes across, the mathematical derivation in Section 4 could be made clearer. One suggestion is to add a bit more step-by-step commentary around Equation (17). Moreover, the authors should include a short explanatory paragraph or a simplified toy example to illustrate how increasing entropy = flattening the reference distribution.

- Broaden experimental evaluation: To strengthen the paper, the authors could evaluate SEE-DPO on additional model (e.g. Stable Diffusion 2.x, SDXL, or PixArt). This will confirm that the entropy regularizer consistently improves alignment and diversity across architectures.

- Enhance figure readability: Redesign the training‑curve plots and figures (e.g. Figure 2, 4, 5) with larger fonts, clearer line styles or markers, and higher-contrast color choices so that key trends and comparisons remain legible even at reduced sizes.

**Strengths And Weaknesses:**

**Strengths**
- Conceptual Insight: The work introduces a regularization term – self-entropy maximization – into the RLHF objective for diffusion models. This idea directly targets the identified problem of reduced diversity. This is a simple yet insightful contribution.

- Soundness: The modified objective (Eq. 17) is derived by augmenting the standard DPO loss with the entropy term. The authors show that this leads to an effective flattening of the reference distribution, which may force the policy/model to be more exploratory. The derivation appears sound and follows the DPO framework closely. Moreover, the approach is straightforward to implement, as it just adds an extra term to the existing KL-regularized DPO objective. It can be easily combined with any DPO-based algorithm without much overhead.

- Performance: The experiments are well-designed to ensure fairness with baselines, cover both objective metrics and human evaluation, and demonstrate improvements due to the proposed method.

**Weaknesses**
- Incremental Novelty: The idea of adding an entropy-based regularization to encourage exploration, while effective, is somewhat incremental. In this paper, the novelty lies in recognizing the importance of such a term for diffusion models and integrating it into the DPO framework. This is a valuable contribution, but also a relatively straightforward extension of existing methods. From a novelty standpoint, it may be seen as an incremental improvement on prior art (D3PO, Diffusion-DPO, etc.) rather than a completely new paradigm.

- Clarity: While the paper provides a correct derivation of the new objective, the theoretical treatment could be clearer. The notation and equations (particularly around Eq. (17)) are quite dense. Readers who are not already familiar with the DPO formulation for diffusion models might struggle to follow this part. The paper could do more to intuitively explain each step of the derivation. For example, the connection between adding the entropy term and effectively raising the reference distribution to a power of $1/(1+\gamma)$ (thus flattening it) is mentioned but not deeply expanded on in the main text. Additionally, some background on how DPO is applied to diffusion (e.g. the role of “winner” and “loser” samples, or how the noise prediction terms relate to preferences) would help make the paper self-contained.

- Limited evaluation scope: The experiments are focused on a single benchmark (Pick-a-Pic V1) and one base model (Stable Diffusion 1.5). It remains uncertain how well SEE-DPO would generalize to other settings. For instance, would the approach yield similar benefits on a different text-to-image dataset or with a different diffusion model (e.g. Stable Diffusion 2.x or another architecture)? This narrow scope means the generality of the conclusions is not fully tested.

- Effectiveness of method on diversity: The qualitative results (Fig.3,  in the paper that aim to show increased diversity do not seem convincing. While the visual quality seems to be improved compared to the baselines, it does not seem to demonstrate increased diversity.

---

> ### Author Response · Authors · 2025-04-22
> **Official Comment by Authors**
>
> Thank you very much for your valuable feedback and the time you dedicated to reviewing our paper. We have revised all graphs and figures to enhance clarity and readability. Due to data loss, we had to rerun the DiffusionDPO experiment shown in Figure 2. Additionally, we have expanded the explanation surrounding Equation 17 and included a small toy experiment in the appendix to further support our claims. Our results are reported using Stable Diffusion v1.5, both due to hardware limitations and to ensure a fair comparison across all methods, as D3PO is also evaluated on Stable Diffusion v1.5. We have also added a border impact section as suggested.

---

### Author Response · Authors · 2025-04-24
**General Comments**

Dear Action Editor and reviewers,

We sincerely appreciate your thoughtful feedback, constructive suggestions, and critical insights. They have been invaluable in helping us refine our work. In response, we have substantially revised the paper to better clarify our contributions and significantly enhance the experimental section. We did rerun the experiments for Diffusion-DPO to generate Figure 2, to support our claims better. These changes are reflected in the updated PDF submitted alongside this response.

If you have any further questions or require additional clarification, we would be more than happy to address them.

Sincerely,
Authors

---

### Decision · Action_Editor_5NjC · 2025-06-04

**Recommendation:** Accept as is

**Comment:**

This paper has several merits, including,
(1) clear problem motivation with broad applicability: the paper tackles a well-recognized limitation in RLHF for diffusion models—namely, model collapse due to overfitting and reward hacking. this problem is relevant and significant across a growing range of generative ai applications.
(2) novelty through self-entropy regularization: the proposed method introduces a self-entropy maximization term into the direct preference optimization (DPO) objective. this encourages broader policy exploration and mitigates over-optimization, resulting in more diverse and stable generation.
(3) theoretical soundness and generalizability:  the authors demonstrate a mathematically sound derivation of the modified loss function and prove equivalence across several DPO variants.

The authors thoughtfully addressed reviewer concerns, notably by re-running key experiments (e.g., Diffusion-DPO), clarifying theoretical derivations (Equation 17), including a toy example, enhancing figure clarity, and so on. Despite one reviewer still has the negative final score, the reviewers largely agreed that SEE-DPO is a meaningful, well-argued contribution. While the entropy regularization idea is not novel in general, applying it systematically and successfully within DPO for diffusion models—while empirically verifying its benefits—is a clear strength.

**Audience:**

The audience includes researchers and practitioners in generative AI, especially those working on diffusion models, RLHF, and preference optimization. It also targets ML methodologists interested in model alignment, regularization, and robustness.

**Claims And Evidence:**

This paper proposes SEE-DPO, a simple yet effective extension to Direct Preference Optimization (DPO) that introduces a self-entropy regularization term to combat overfitting and reward hacking in diffusion models. The method encourages more diverse and robust generations by promoting exploration. Experiments on Stable Diffusion show improved image quality, diversity, and training stability across multiple DPO variants.

It has  strong experimental evidence across key metrics. Despite limited compute, the paper demonstrates consistent improvements in both diversity and alignment quality, reporting substantial gains in metrics such as RMSE, SSIM, PSNR, and ImageReward. Importantly, these results are supported by retrained baselines and improved visualizations post-revision.